**communications** engineering

# Genetically synthesized supergain broadband wire-bundle antenna
Dmytro Vovchuk [1,3] ✉, Gilad Uziel[1,3], Andrey Machnev[1], Jurgis Porins[2], Vjaceslavs Bobrovs [2] & Pavel Ginzburg[1]

High-gain antennas are essential hardware devices, powering numerous daily applications, including distant point-to-point communications, safety radars, and many others. While a common approach to elevate gain is to enlarge an antenna aperture, highly resonant subwavelength structures can potentially grant high gain performances. The Chu-Harrington limit is a standard criterion to assess electrically small structures and those surpassing it are called superdirective. Supergain is obtained in a case when internal losses are mitigated, and an antenna is matched to radiation, though typically in a very narrow frequency band. Here we develop a concept of a spectrally overlapping resonant cascading, where tailored multipole hierarchy grants both high gain and sufficient operational bandwidth. Our architecture is based on a near-field coupled wire bundle. Genetic optimization, constraining both gain and bandwidth, is applied on a 24-dimensional space and predicts 8.81 dBi realized gain within a half-wavelength in a cube volume. The experimental gain is 8.22 dBi with 13% fractional bandwidth. The developed approach can be applied across other frequency bands, where miniaturization of wireless devices is highly demanded.

Antenna elements are essential hardware to enable wireless communication links. Among many different characteristics of those devices, gain plays an important role as it has a primary impact on link budget calculations, used to estimate maximally available distances to maintain a connection[1].

There are quite a few antenna architectures, developed to achieve high gain. A specific choice of layout depends on frequency range and other engineering application-specific parameters, constraining the design. In a broader sense, an antenna operation principle can be separated into a pair of categories, i.e., resonant and nonresonant. The latter class, being preferable in terms of bandwidth, scalability, integrability within circuitry, material compatibility, and several other parameters, trades gain properties for a physical aperture size. A typical example here is parabolic reflectors or horns, whose gain is directly proportional to the aperture area (A), normalized to an operational wavelength squared ($\lambda^2$)[1]. Resonant antennas, on the other hand, can have a rather small physical aperture (e.g., dipole) and can encompass many resonant elements within a design. Typically, antenna size is assessed versus a radius of a virtual sphere (R), enclosing the structure. Theoretically speaking, electrically small structures with R smaller than $\lambda$ (also, $2R/\lambda < 1$ might be used as a criterion) can possess high gain properties[2]. In this case, multipolar resonances within a structure should interfere constructively with balanced amplitudes and phases. It is worth noting, that radiation patterns are better described by spherical harmonics (multipoles), forming a complete set of functions on a sphere[3]. However, in structures lacking rotational symmetry, internal resonances are not 1:1 mapped on the multipole expansion, forming the far-field radiation[4–7]. Nevertheless, it is quite clear that several higher-order multipoles should be employed in superdirective antenna design[5,7]. As a result, the structure becomes extremely susceptible to fabrication imperfections, the surrounding environment, and ohmic losses within constitutive materials – all those owing to a resonant near-field accumulation within the device. While those aspects are very well known[2], the challenge is to find architectures that are (i) subject to fast optimization, (ii) experimentally realizable, (iii) based on low-loss materials, (iv) can be matched to radiation without employing lossy lumped elements, and (v) provide sufficient operational bandwidth.

Several superdirective antennas were demonstrated, including architectures, based on ceramic resonators (experimentally)[8,9], multilayer designs (theory)[10–13], Huygens sources[14–19] (though with a relatively small number of low-order multipoles + also tuned with lumped elements[5,20–24], and high-impedance surface antennas[25]. Small antenna arrays, with basic elements including, patches, PIFA, monopoles, and others are used for implementing superdirective devices. However, in many cases, an infinite-size (in theory) ground plane serves as an integral part of the system, questioning the validity of applying an electrically small criterion[23]. The introduction of lumped elements into the design, being a promising approach for resonant structures miniaturization, results in elevated losses. Thus, the realized gain of devices drops, nevertheless the directivity can be quite high[22]. A promising

[1]School of Electrical Engineering, Tel Aviv University, Ramat Aviv, Tel Aviv, Israel. [2]Institute of Photonics, Electronics and Telecommunications, Riga Technical University, Riga, Latvia. [3]These authors contributed equally: Dmytro Vovchuk, Gilad Uziel. ✉e-mail: dimavovchuk@gmail.com

platform, in application to superdirectivity and superscattering, is small arrays of near-field coupled resonators[26–31], optimized with the aid of fast genetic algorithms.

Here we explore a wire bundle architecture to optimize an antenna resonator for achieving broadband supergain performance. This architecture complies with the five following (also discussed above) requirements, essential for a practical realization of supergain devices. (i) Efficient optimization - an array of vertically aligned wires is taken as a starting point and then optimized to maximize antenna gain and bandwidth while keeping its footprint small. Each element in the array is allowed to move within the volume and has a different length, though a tilt is not included. Those degrees of freedom form a search space. (ii, iii) The realization is based on copper wires, pinched into a Styrofoam. Near-fields are accumulated in a free space between the resonators, making the structure less susceptible to losses in practice. (iv, v) Resonant cascading of interfering multipoles is designed to provide a matching and high directivity over a broad frequency range.

The manuscript is organized as follows: genetic optimization of the structure is presented first and then followed by a detailed electromagnetic analysis to reveal its operation principles. Experimental realization and antenna characterization, along with a comparison to several well-established limits, come before the Conclusion.

## Results

### Antenna design

To obtain resonant multipoles overlapping in a subwavelength structure, simultaneous optimization over many degrees of freedom is required. A direct search, in this case, will result in exponential growth of computational resources, motivating one to apply different optimization strategies. Evolutionary algorithms, with genetic optimization being a subset, is a possible compromise, which is intensively explored to design various functional structures. The concept is to treat an electromagnetic configuration as a basic provision in the theory of biological evolution, where processes of selection, mutation, and reproduction govern future development. Evolutionary algorithms[29,32–34] are widely used in multi-dimensional domains, where the functional dependence between parameters is either non-differentiable or has many local extrema. Genetic optimization also allows for handling multi-objective problems, providing a set of solutions that offer different trade-offs among the objectives, rather than a single optimal solution. Genetic optimization also has many drawbacks, e.g., convergence to optimal solutions is not guaranteed. These algorithmic approaches were introduced into engineering problems in the 1960–70s[35] and since then, being supported by ever-growing computational power, started to shift aside from conventional design rules, e.g.[36,37],. In the context of this report, it is worth mentioning existing studies looking into circular rods superscatterers[38], superabsorptive nanoparticles[39], core-shell cylindrical superscatterers[40], subwavelength superscattering nanospheres[41], antenna design, nanoplasmonic particles[30,42,43], and others[44–50].

Before performing an optimization, we will set up our search space. The structure encompasses 9 vertically aligned metal wires. One of them is an active radiating element, which is set fixed. As a standalone, it performs as a dipole antenna, matched to radiate at 6 GHz. The rest 8 elements, being initially positioned at the nodes of a symmetric $3 \times 3$ array, are allowed to change their position and length independently of each other. This forms an $8 \times 3 = 24$-dimensional search space. In this realization, the wires are kept mutually parallel, and tilt is not allowed. The algorithm flow and the schematic layout of the structure appear in Fig. 1. In brief, optimization is based on the principles of Genetics and Natural Selection. A population of possible solutions repeatedly undergoes recombination and mutation, producing new children. Each realization is assigned versus a pre-defined fitness function, and the best (at this stage) individuals are given a higher chance to evolve. This process keeps repeating until reaching a stopping criterion. Here, an upper limit of 1000 iterations was chosen. The main parts of the algorithm, summarized in the chart, include (i) Selection – selecting individuals, called parents, that contribute to the population of the next

generation. The selection is generally stochastic and can depend on the individuals' scores (gain properties here), (ii) Crossover – combining two parents to form children for the next generation, (iii) Mutation – applying random changes to individual parents to form children. The choice of the fitness function requires extra care, as it has a major impact on the convergence. While the gain and bandwidth can be constrained within the same cost function (e.g., maximizing their product ($G B$), weighted sum ($a1G + a2B$), or a similar figure of merit), this approach was empirically found less efficient. Our reasoning is based on the observation that the highly resonant structure we are examining is extremely sensitive to changes, a characteristic common in superscattering and superdirective designs. Therefore, imposing too many constraints at once can result in instability. Consequently, the optimization was split into two stages – the gain was optimized first (to surpass the Chu-Harrington limit and tuned manually above this value) and only those species reaching the threshold were further promoted to the gain assessment (aimed at 10% fractional bandwidth, relevant to wireless communication needs) (Fig. 1a). The bandwidth of superdirective performance was determined by identifying the frequency range where the gain exceeds the Chu-Harrington or Geyi's limits (refer to Fig. 1a). Furthermore, it was verified that within this range, the gain remains continuously above the specified limit without any drop. In addition, worth mentioning more advanced approaches, i.e., meta-learning[51,52], which can optimize gain-bandwidth cost function to be used in optimization.

The layout of the optimized structure, which was chosen for the subsequent realization, appears in Table 1. Rows correspond to the wire length and two coordinates within a polar system, linked to the center of the initial array (Fig. 1b, c). For convenience, the central element is also parametrized, using this approach. Nevertheless, its 2 degrees of freedom (r, φ) are not independent, as the entire geometry can be linearly shifted and rotated to keep it at the center. The numbering of elements is indicated in Fig. 1c, where 'm' stays for the center (middle) element and 0 is the feed. Since the genetic algorithm does not ensure reaching a global extremum, it can provide other equally valid (according to the threshold conditions) solutions. Two other layouts, demonstrating similar performance, are provided in the Supplementary Note 1.

### Numerical analysis and experimental verification

To assess the performance of the optimized model, detailed numerical and experimental characterizations will be made next. CST Microwave Studio (Frequency Domain solver) was used to perform the antenna analysis. Experimental measurements were performed with the setup (Fig. 2a) that consists of the receiving broadband IDPH-2018 S/N-0807202 horn antenna certified in the frequency range of 2–20 GHz and the 360° rotating table with the foam stand to avoid unwished reflections. The antenna under test (AUT) is located in the middle of the rotating table.

The experimental realization of the AUT is based on copper cut wires, placed in a Styrofoam holder. Figure 2b, c presents photographs from different perspectives. To re-emphasize, the choice of the material platform here is quite important, as superdirective-supergain structures are susceptible to material losses and fabrication tolerances. Therefore, Styrofoam is selected as a host material due to its transparency for electromagnetic waves in the microwave frequency range.

The sample was fabricated by pinching copper wires into the Styrofoam host. The achieved fabrication accuracies are ±0.03 mm of the wire length and ±0.01 mm in wire positioning, which were obtained by printing the array template on a transparent slide and processing the layout under a table-top microscope. The feeding element is a conventional dipole antenna with λ/4-balun.

Figure 2d shows the S11 parameter as a function of frequency. A rather good impedance matching (−25 dB around 6 GHz) is observed and predicted numerically. The rest of the panels in Fig. 2 demonstrate the numerically obtained 3D radiation pattern (Fig. 2e) and azimuthal and elevation cuts for both numerical and experimental data (panels f and g). Highly-directive shapes are observed.

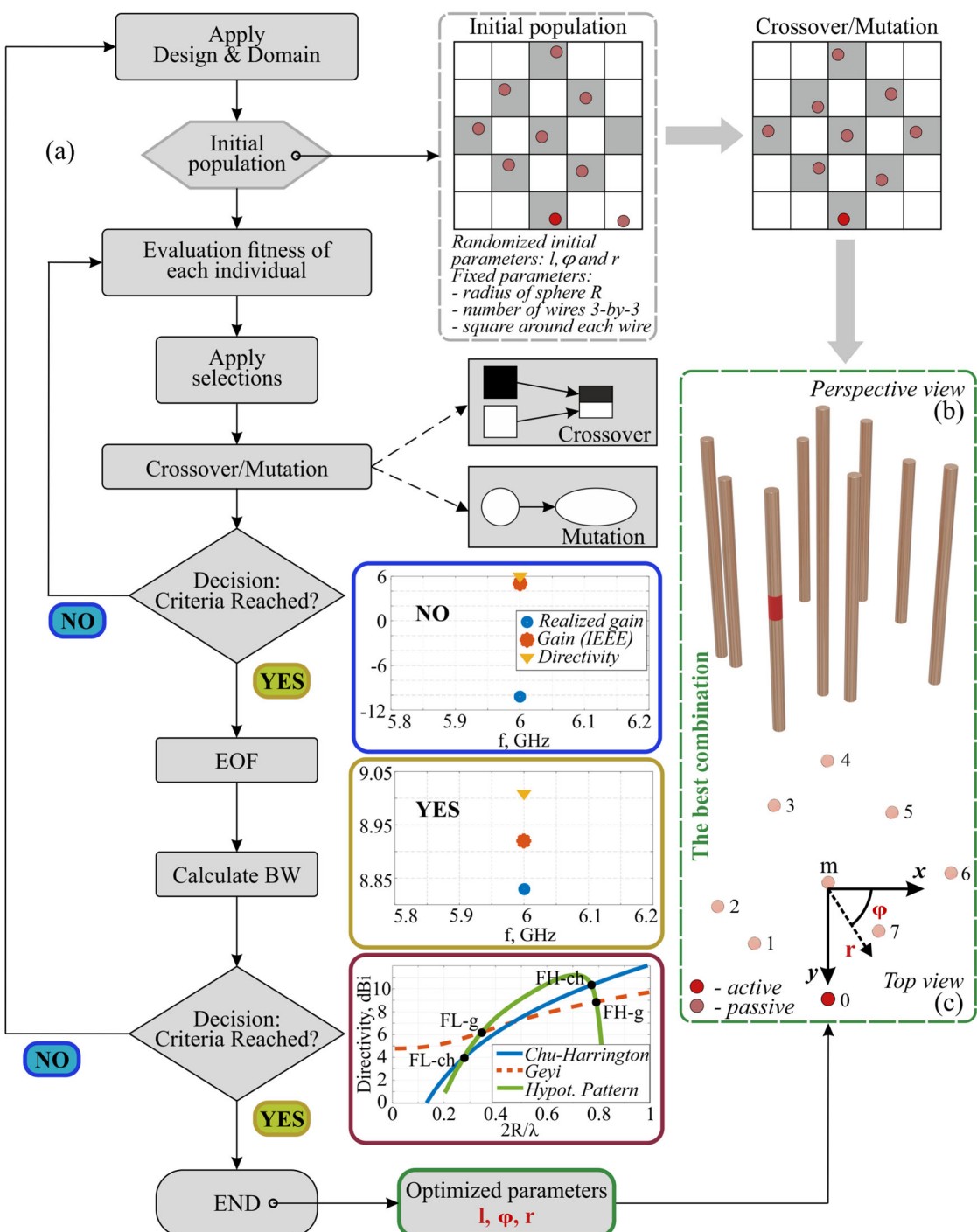

**Fig. 1 | Optimization algorithm. a** A flow chart of the superdirective-supergain antenna optimization with a genetic algorithm. **b, c** the structure under optimization – perspective and top view, respectively. Optimization degrees of freedom are indicated in **c**. The gray squares show the bounds, restricting the wires' possible positions. A red inset in a wire is a feeding port.

To characterize the results quantitatively, the following definitions are in use[1,53]:

$$D_\varphi = \frac{P_{\max}}{\frac{1}{2\pi}\int_0^{2\pi} P(\varphi)\,d\varphi} \tag{1.1}$$

$$D_\theta = \frac{P_{\max}}{\frac{1}{2\pi}\int_0^{2\pi} P(\theta)\,d\theta} \tag{1.2}$$

$$D = \frac{P_{\max}(\varphi) + P_{\max}(\theta)}{\frac{1}{4\pi}\left[\int_0^{2\pi} P(\varphi)\,d\varphi + \int_0^{2\pi} P(\theta)\,d\theta\right]} \tag{1.3}$$

where $D_\varphi$ and $D\theta$ stay for directives in azimuthal and elevation cut planes, while $D$ is the total directivity of the antenna. P is the far-field radiated power density. Table 2 summarizes the results. For an assessment, 9 dBi directivity can be achieved with an electrically large Yagi-Uda antenna, encompassing 6-7 elements and having a smaller

https://doi.org/10.1038/s44172-024-00235-y   **Article**

**Table 1 | The parameters of the optimized antenna, the numbering corresponds to Fig. 1c**

|           | Center (m) | Feed (0) | 1     | 2      | 3     | 4     | 5     | 6     | 7     |
|-----------|-----------|----------|-------|--------|-------|-------|-------|-------|-------|
| $l$, mm   | 28.16     | 22.32    | 18.23 | 21.6   | 16.67 | 18.7  | 16.2  | 20.91 | 21.46 |
| $r$, mm   | −0.35     | 8.48     | 9.4   | 8.51   | 7.44  | 9.61  | 7.46  | 9.41  | 5.09  |
| $\varphi$, deg. | 0   | 0        | 163   | 170.4  | 237   | 270   | 310   | 353   | 41.25 |

**Fig. 2 | Numerical and experimental results for the antenna characterization. a** Experimental setup for measurements of antenna's characteristics. **b**, **c** Photographs of the antenna under test (AUT) from different perspectives. **d** S11-parameter spectra - numerical (red) and experimental (blue) curves. **e** 3D radiation pattern, numerical result. **f**, **g** Radiation patterns in azimuthal (green) and elevation (yellow) cuts – numerical and experimental results. Patterns are plotted for 6 GHz, where the antenna directivity is the highest.

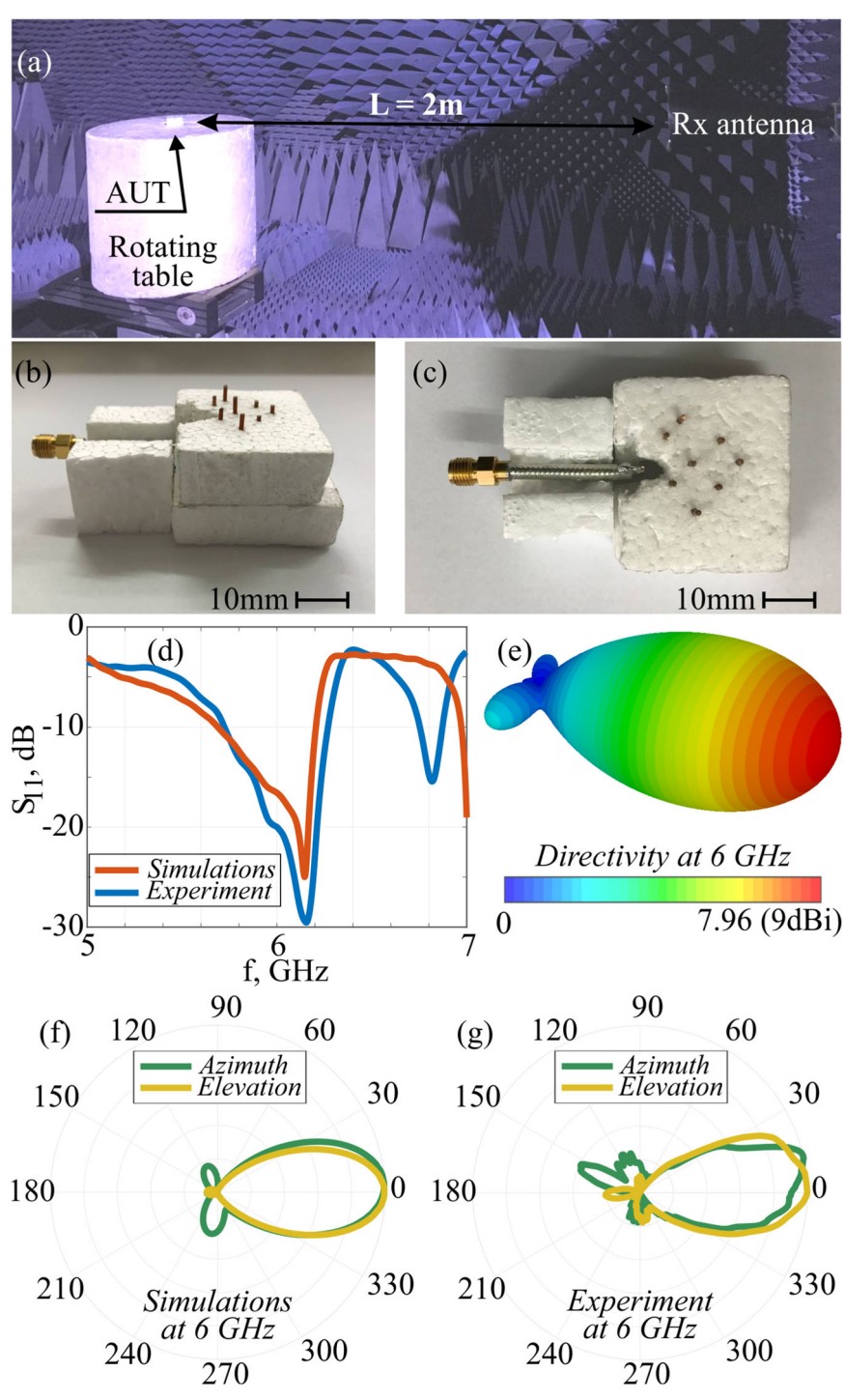

bandwidth – thus our structure operates differently and belongs to a different class (small, supergain) of antennas. It is also worth noting an approach to shaping antenna patterns by creating plasma-based reflectors, which can be switched on and off on demand, e.g. refs. 54–58.

Apart from the directivity, the realized gain has been measured and the result appears in Table 2. To perform the calibrated measurements, the same IDPH-2018 S/N-0807202 horn antenna was used as an etalon – it was placed on the rotating table and measured in its two principal planes. The measurements were assessed versus the tabulated data and normalization

**Table 2 | Directivities and gain – numerical and experimental results at 6 GHz**

|  | Numerical model | Experiment |
|---|---|---|
| $D_\varphi$, dBi | 5.95 | 6.44 |
| $D0$, dBi | 7.39 | 5.54 |
| $D$, dBi | 9.01 | 9.15 |
| $G$, dBi | 8.81 | 8.22 |

factors have been extracted. The gain calculations were made by processing the experimental data, obtained at 2 principal planes. It is worth noting that identifying those might be quite challenging if AUT has multiple lobes. In this case, the full characterization required a complete $4\pi$ scanning. In our case, however, a single well-pronounced lobe is predicted numerically. Furthermore, we manually checked it by scanning the antenna at a tilt. Given a single main lobe, the gain $G$ is calculated from the orthogonal $G_\varphi$ and $G\theta$ components (Eq. 2.1)[1,53]. Those are extracted from S-parameters, according to Friis principle. AUT's gain is extracted as follows:

$$G = G_\varphi + G_\theta, \quad (2.1)$$

$$P_r = P_t G_r G_t \left(\frac{\lambda}{4\pi L}\right)^2 \quad (2.2)$$

$$G_r G_t = \left(\frac{\lambda}{4\pi L}\right)^2 \frac{P_t}{P_r} \quad (2.3)$$

$$G_{AUT} = G_t = \left(\frac{\lambda}{4\pi L}\right)^2 \frac{S_{21}^2}{G_r} \quad (2.4)$$

where $P_r$ and $Pt$ are power values of the same receive and transmit antennas with the known gain values $Gr$ and $Gt$; $L$ is the distance between antenna apertures; and $\lambda$ is a wavelength. Equation 2.2 allows performing the appropriate calibration of the measurement system via Eq. 2.3, where $Pt/Pr = (S21)^2$ is measured with a network analyzer. Then a gain of the AUT is extracted following Eq. 2.4.

Owing to an accurate alignment of wires, the antenna has a well-defined linearly polarized radiation pattern (as a dipolar antenna has). Cross-polarization effects above a noise level in the measurements were not observed.

To reveal the radiation characteristics of the antenna, its directivity and gain spectra are plotted (Fig. 3a, b, respectively). It is quite remarkable that the genetic algorithm allowed the construction of the antenna with a sufficiently large operational bandwidth. It is a result of two main contributing factors. First, the antenna, nevertheless is electrically small, it is not deeply subwavelength. The second effect comes from the resonance cascading approach, where multiple resonances of the system do have a spectral overlap but cover a sufficient bandwidth owing to their side-by-side mutual positioning. This effect will be analyzed next.

**Multipole expansion**

Superdirectivity properties come from constructive interference of several resonant multipoles, which sum up into a directive radiation pattern. To demonstrate the effect, the multipole expansion of the radiation pattern will be performed next. There are a few possible approaches for analyzing electromagnetic scattering and radiation problems in various systems, e.g. refs. 59–70. Figure 4 demonstrates the spectra of several lower-order multipoles[7]. The convergence of the multipolar sum to the total radiated power is assessed. First, it can be seen that the six lowest multipoles are sufficient to describe the antenna radiation. In previous reports, only a few lower-order multipoles were considered to create a directive pattern, thus lower values have been demonstrated and the bandwidth was not assessed[18].

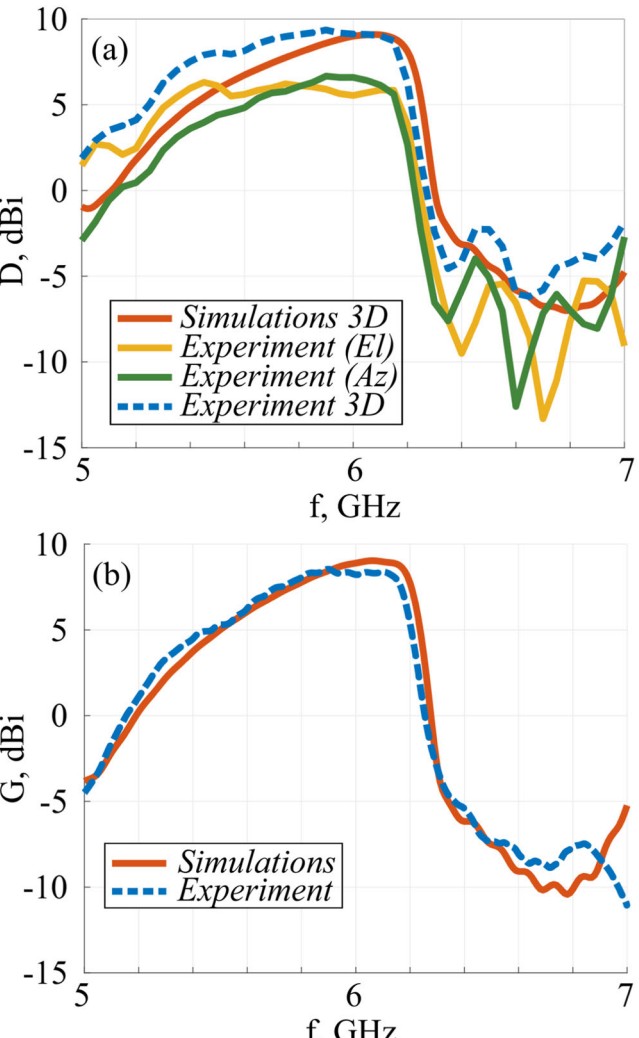

**Fig. 3 | Directivity and gain assessment. a** Azimuthal, elevation, and total directivity spectra – numerical and experimental results. **b** Gain spectrum – numerical (red) and experimental (blue).

Second, the multipoles resonate at the vicinity of 6 GHz and cover a 13% fractional bandwidth, as the resonances are placed side by side. This resonant cascading approach to bandwidth extension allows superdirective antennas to support wireless links within existing communication standards and motivates applying the proposed methodology to miniaturization. Overall, multipolar engineering is an essential tool for the design and analysis of compact directive devices[6,7,13].

**Performance assessment**

After revealing the antenna performances and underlining the principles of its basic operation, the device can be assessed versus existing and widely used bounds and compared with other superdirective demonstrations. The assessment will be made versus Chu-Harrington and Geyi limits, being the most commonly used ones and straightforward to apply, though others have been developed, e.g.[5,71,72].

While Chu-Harrington and Geyi formulas are considered for initial assessment criteria, those are not fundamental limits. In this context, it is worth mentioning recent advanced tools, based on convex optimization, which allow assessing fundamental upper limit on a certain set of performances, e.g., a gain at a given frequency[5,71]. However, applying powerful inverse algorithm methods (e.g.[73]) still requires introducing topological constraints and are not generically applicable to a variety of antennas,

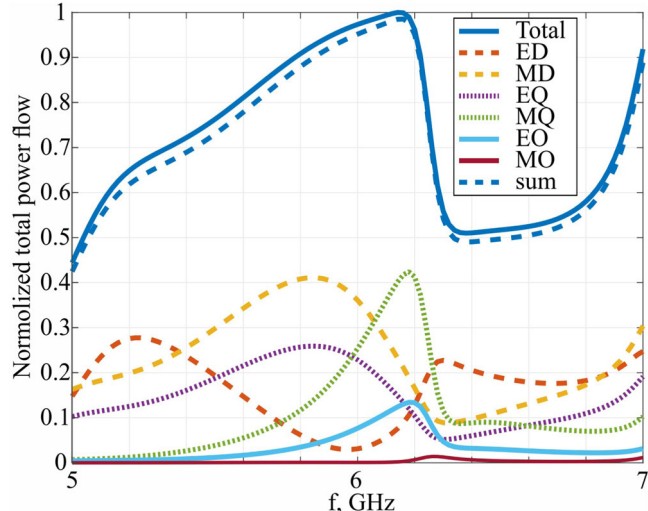

**Fig. 4 | Multipole expansion of the antenna radiation pattern.** ED, MD – electrical and magnetic dipoles, EQ, MQ – quadrupoles and EO, MO – octupoles contributions.

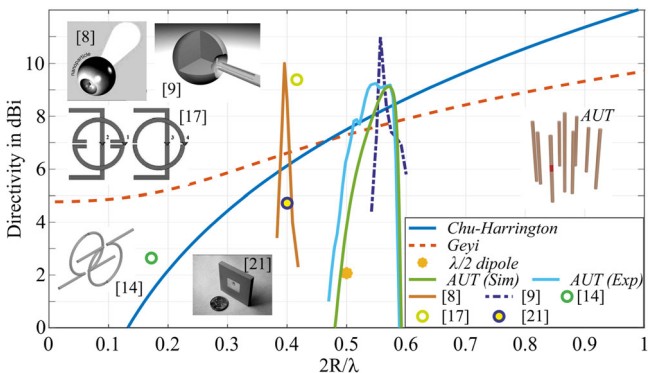

**Fig. 5 | Chu-Harrington and Geyi limits[74] versus the electrical size of an antenna.** Points on the chart correspond to several reported realizations, where performances at a single frequency were reported (references are in the legend). Solid lines – directivity spectra, calculated from data, adopted from indicated references. Antenna under test (AUT) – our realization with both numerical and experimental results presented—yellow dot - conventional λ/2 dipole antenna for reference.

### Table 3 | The bandwidth of the superdirective broadband antennas over the fundamental limits (Fig. 5)

| Ref. | %BW, over Chu-Har. | %BW, over Geyi |
|---|---|---|
| AUT(Sim.) | 6.6 | 8.3 |
| AUT(Exp.) | 12.2 | 13 |
| Capek & Jelinek[5] | 4.4 | 3.8 |
| Ziolkowski[6] | 2.7 | 3.7 |

calculated operational bandwidth. Specifically, gain spectra were calculated and plotted versus 2 R/λ, the x-axis parameter in Fig. 5. Solid line in Fig. 5 corresponds to the calculation. In the case of our design, directivity spectra (both numerical and experimental) are presented. According to the figure of merit, related to the bandwidth, our antenna outperforms its counterparts, as demonstrated in the chart.

To summarize the bandwidth performances quantitatively, the superdirective bandwidth was defined versus both Chu-Harrington and Geyi limits – the frequency for which the directivity is above those bounds. Table 3 summarizes the results. Fractional superdirective bandwidth is calculated as the ratio between a range where the gain is superdirective over the central frequency. According to this definition, our realization has an advantage over existing reports. Furthermore, the demonstrated numbers make the device a legitimate hardware to support wireless communication links.

Several other superdirective supergain structures, resulting from our genetic algorithm appear in Supplementary Note 1 and Supplementary Fig. 1a–f.

We simulated these structures and showed their directivity D, realized gain G, and calculated efficiency as E = G/D that are shown in Supplementary Note 1 Supplementary Fig. 1g–i, respectively. The analyzed multipole decomposition (Supplementary Note 1 and Supplementary Fig. 1j) and directivity over Chu-Harrington and Geyi limits (Supplementary Note 1 and Supplementary Fig. 1k, l) confirm that the algorithm gives other broadband superdirective layouts, however the considered main structure possesses the best characteristics.

## Conclusion
A supergain broadband antenna has been demonstrated. The concept of resonant cascading has been developed and employed towards achieving both high directivity and bandwidth operation within a subwavelength device. While a spectral collocation of many resonant multipoles is responsible for obtaining highly directive patterns, tuning multipole hierarchy and spreading their resonant responses over a span of frequencies can grant both directivity and bandwidth. These generally competing parameters can establish a Pareto front[78], which could be explored in future investigations. Here, we demonstrate a compact, electrically small antenna with 9dBi directivity and 13% fractional bandwidth at 6 GHz. Resonant cascading of six lower-order multipoles was shown to govern the radiation characteristics. The experimentally demonstrated gain is 8.22 dBi.

The capability to achieve high gain properties within a small footprint device and without bandwidth degradation makes the proposed superdirective elements very attractive in many wireless applications, owing to new strategies in hardware elements miniaturization.

## Methods
Electromagnetic design and simulations were performed in CST Microwave Studio.

Measurements were performed in an anechoic chamber certified for the frequency range 1-20 GHz that is shown in Fig. 2a. The measurement system MIDAS that includes 360° rotation table and scanning device was installed by ORBIT/FR Engineering Ltd. The broadband antenna (2–20 GHz) NATO IDPH-2018 horn was used as Rx antenna (Fig. 2a). The AUT was precisely fabricated in RF laboratory with the appropriate tools.

considered in practical applications (e.g., antennas, encompassed in large-scale electromagnetic systems). Thus, Chu-Harrington and Geyi limits remain widely used, generic, and easy to apply assessment criteria. It's important to clarify that the bounds mentioned earlier do not define the actual shape of the antenna. Rather, they provide a theoretical prediction of the maximum potential performance of such an antenna. In contrast, topology and inverse optimization algorithms are capable of delivering tangible designs.

Figure 5 is the directivity versus an electrical size of device size (2R/λ, where R is the radius of an enclosing sphere and λ is the operational wavelength). Several recently reported superdirective antennas were chosen for making the assessment[74–77]. It is worth re-emphasizing those designs, miniaturized with lumped elements, do not demonstrate high gain characteristics and are thus omitted from the discussion. Our lumped-element-free device is well positioned above the Chu-Harrington and Geyi limits. The dots on the chart correspond to the values reported in the literature. It is worth noting that in a vast majority of cases, bandwidth characteristics are not discussed as it is widely believed that a superdirective antenna must be narrowband (we question this claim by demonstrating our architecture). To address the gain properties, we explored several reported designs and

The AUT (Fig. 2b, c) and Rx antenna were connected to N5232B PNA-L Microwave Network Analyzer 300 kHz– 20 GHz (PNA—Performance Network Analyzer) which was automatically controlled by MIDAS.

Post signal-processing, data analysis and calculations were done with MatLab.

## Data availability

The data that support the findings of this study are available from the corresponding author upon reasonable request.

## Code availability

The codes used in this work are available from the corresponding author upon request.

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

## Acknowledgements

D.V., G.U., A.M., and P.G. acknowledge the support from the Department of the Navy, Office of Naval Research Global under ONRG award number N6290921-1-2038 and Israel Science Foundation (ISF grant number 1115/23) J.P. and V.B. acknowledge the support from the RRF project Latvian Quantum Technologies Initiative Nr. 2.3.1.1.i.0/1/22/I/CFLA/001. The project was supported, in part, by ERC POC "DeepSight".

## Authors contributions

D.V. was responsible for the design and simulations, experimental fabrication, measurements, data analysis, and characterization. G.U. – optimization and design, simulations. A.M. was responsible for multipole decomposition and data analysis. V.B. and J.P. participated in research and discussions. P.G. supervised the project and revised the paper; author of idea. All the authors contributed to discussions and writing of the manuscript.

## Competing interests

The authors declare no competing interests.
