## [Peer Review File · Communications Engineering]

Reviewers' comments:

Reviewer #2 (Remarks to the Author):

The area of super-directive (super-gain) antennas and antenna arrays is well covered in the literature. This is well reflected by the list of references provided by the authors. I think, however, that some important works are missing, especially those closely related to the topic of the manuscript.

When designing a new device, two questions must be addressed subsequently:

1. What are the performance limits of a hypothetical device?
2. How do we approach these limits?

As for the first question, the authors used the classical bounds of Harrington and the bounds proposed by Geyi. They both are rendered obsolete. Recent works determining bounds on antenna gain are published [1], and should be used to estimate how distant the actual antenna design is from principal bounds and to be compared with Harrington's bound, see, e.g., [2] as well. The only question is the identification of the optimal shape/material arrangement. This is the subject of the second question. . .

The dipole arrays are studied in many papers, e.g., in [3] or [4]. The achieved 8.8 dBi gain can equivalently be realized with only three dipoles, see [5]. Why to use so many of them? The authors should study diminishing returns when using 3-4-5- . . . dipoles.

I have the following questions and concerns:

- ☐ Why did the authors use a heuristic algorithm? Why genetics?
- ☐ Topology optimization is commonly applied to similar problems. Why is it not used here?
- ☐ "Nevertheless, it is quite clear that several higher-order resonances should be employed in super-directive antenna design." It is not quite clear to me. What does it mean "higher-order resonance"? Does it mean that higher-order (spherical) modes are used too and tuned to resonance?
- ☐ It is not obvious how exactly is the fitness function composed. The authors mentioned they simultaneously optimized for gain and bandwidth. How is it done? What is the weight between these metrics?
- ☐ Multi-pole expansion discussion is interesting and similar to many existing papers, see [6, 7, 8]. However much more precise modal analysis is possible based on characteristic mode decomposition [9].

The procedure proposed by the authors relies on a genetic algorithm. Each run will produce a different antenna. Each frequency or different number of radiators will also produce remarkably different results. As such, the reviewer does not see the true value for the reader. What information can be distilled from this contribution to be applicable in general? What can we learn from that as a general design strategy to adopt when the conditions are altered?

Another problem is that the proposed design does not reach the advanced level of other contemporary designs, mostly published within Antenna & Propagation Society's journals.

References

- [1] M. Gustafsson and M. Capek, "Maximum gain, effective area, and directivity," *IEEE Trans. Antennas Propag.*, vol. 67, pp. 5282 – 5293, Aug. 2019.
- [2] P.-S. Kildal, E. Martini, and S. Maci, "Degrees of freedom and maximum directivity of antennas: A bound on maximum directivity of nonsuperreactive antennas.," *IEEE Antennas and Propagation Magazine*, vol. 59, no. 4, pp. 16–25, 2017.
- [3] A. Clemente, M. Pigeon, L. Rudant, and C. Delaveaud, "Design of a super directive four-element compact antenna array using spherical wave expansion," *IEEE Transactions on Antennas and Propagation*, vol. 63, no. 11, pp. 4715–4722, 2015.
- [4] H. Jaafar, S. Collardey, and A. Sharaiha, "Characteristic modes approach to design compact superdirective array with enhanced bandwidth," *IEEE Transactions on Antennas and Propagation*, vol. 66, no. 12, pp. 6986–6996, 2018.
- [5] A. Debard, A. Clemente, A. Tornese, and C. Delaveaud, "On the maximum end-fire directivity of compact antenna arrays based on electrical dipoles and huygens sources," *IEEE Transactions on Antennas and Propagation*, vol. 71, no. 1, pp. 299–308, 2023.
- [6] R. W. Ziolkowski, "Mixtures of multipoles—should they be in your EM toolbox?," *IEEE Open Journal of Antennas and Propagation*, vol. 3, pp. 154–188, 2022.
- [7] R. W. Ziolkowski, "Using huygens multipole arrays to realize unidirectional needle-like radiation," *Physical Review X*, vol. 7, no. 3, p. 031017, 2017.
- [8] R. W. Ziolkowski, "Superdirective unidirectional mixed-multipole antennas: Designs, analysis and simulations," *IEEE Trans. Antennas Propag.*, pp. 1–1, 2023.
- [9] M. Capek and L. Jelinek, "The upper bound on antenna gain and its feasibility as a sum of characteristic gains," *IEEE Transactions on Antennas and Propagation*, pp. 1–1, 2023.

Reviewer #3 (Remarks to the Author):

The paper addresses the design of a super-directive antenna, made of an array of cylindrical scatterers (referred to as wire-bundle here). The paper requires major revisions and clarifications to be suitable for the publication. I have several comments as follows:

1. The idea of using several scatterers to design such directive antennas has been known to the EM community for a long time. There are several papers in this regard. Here, I just refer to one:

Wu, Xiao Po, Jia-Ming Shi, Zong Sheng Chen, and Bo Xu. "A new plasma antenna of beam-forming." *Progress In Electromagnetics Research* 126 (2012): 539-553.

It would be great if the authors could emphasize on the novelty of their work.

2. The authors mentioned in the abstract that:

"The experimental gain is 8.22 dBi with 13% fractional bandwidth, which is the best performance in the field of superdirective antennas now."

I would suggest removing non-scientific terms such as "best" here without any specific quantification.

3. The authors claim that:

"The developed approach can be applied to low-frequency (e.g., kHz-MHz) applications"

It is not then clear why their antenna design is at 6 GHz. It would be great if they could clarify it.

4. What is the numerical solver used in the CST? The time domain, the frequency domain, or any other ones?

5. What is the biggest size of the antenna? Based on the data shown in Table 1 (considering Fig. 1(c)), it seems that the distance between 4 and 0 can be a good approximation in this regard. The distance is almost $9.61+8.48=18.09$ mm, which is equivalent to 0.36λ at 6 GHz. Is it a very small antenna?

Reviewer #1 (Remarks to the Author):

The paper addresses the design of a super-directive antenna, made of an array of cylindrical scatterers (referred to as wire-bundle here). The paper requires major revisions and clarifications to be suitable for the publication. I have several comments as follows:

1. The idea of using several scatterers to design such directive antennas has been known to the EM community for a long time. There are several papers in this regard. Here, I just refer to one:

Wu, Xiao Po, Jia-Ming Shi, Zong Sheng Chen, and Bo Xu. "A new plasma antenna of beam-forming." *Progress In Electromagnetics Research* 126 (2012): 539-553.

It would be great if the authors could emphasize on the novelty of their work.

Our Response:

We absolutely agree with the Referee that directors and reflectors are commonly used by the antenna community to shape radiation patterns. Following the recommendation, we enlarged the discussion on this part. The whole point is how to design structures to get simultaneously (i) a small antenna, high-gain (not only directivity), and (ii) broadband operation. Our report addressed those aspects simultaneously.

The second point is the new class of antennas, based on plasma reflectors – the Referee has provided a very interesting reference (cited now + several others) in this new direction. We admit we overlooked this new approach to beamforming, which we included in the revised version. Worth noting though that this method still requires developing designs for the 3 specs we mentioned above. We touched upon all those in the revised version.

2. The authors mentioned in the abstract that:

"The experimental gain is 8.22 dBi with 13% fractional bandwidth, which is the best performance in the field of superdirective antennas now."

I would suggest removing non-scientific terms such as "best" here without any specific quantification.

Our Response:

We agree and remove the non-quantitative assessment.

3. The authors claim that:

"The developed approach can be applied to low-frequency (e.g., kHz-MHz) applications"

It is not then clear why their antenna design is at 6 GHz. It would be great if they could clarify it.

Our Response:

The general motivation is that miniaturization is especially important for low-frequency applications, where half-lambda is a physically big quantity.

We agree with the Referee – to prevent confusion (since we didn't really demonstrate any low-frequency calculation/experiment), we removed the claim from the abstract.

4. What is the numerical solver used in the CST? The time domain, the frequency domain, or any other ones?

Our Response:

The simulations were initially performed with the Frequency Domain and then also checked with the Time Domain solver – we added those details in the revised version.

5. What is the biggest size of the antenna? Based on the data shown in Table 1 (considering Fig. 1(c)), it seems that the distance between 4 and 0 can be a good approximation in this regard. The distance is almost $9.61+8.48=18.09$ mm, which is equivalent to 0.36λ at 6 GHz. Is it a very small antenna?

Our Response:

Yes, this is indeed a small antenna. For assessing the size, we typically consider the smallest virtual sphere which encompasses the structure. Here, the radius is 28.16 mm (~half-wavelength) is larger than the distance between the wires 0 and 4. Following the recommendation, we included a clarification on this point in the revised.

Reviewers' comments:

Reviewer #2 (Remarks to the Author):

The paper has been improved. There are, however, still some issues that might be corrected by the authors to make the manuscript better.

1/ The authors mixed the bounds (Harrington, Geyi), other bounds (Gustafsson, Capek, et al.), and inverse design (topology optimization). The bounds (Harrington, Geyi, Gustafsson, Capek, et al.) do not propose the actual shape of the antenna. They only predict what can, in principle, be the maximum value of such an antenna. Meanwhile, topology optimization/inverse optimization algorithms can deliver actual design. Notice that convex optimization routines used by Gustafsson and Capek also do not deliver an antenna, only establishing bounds, which is, as compared to Harrington/Geyi, the tight bound. Based on this, I believe the text on page 6, right column, must be slightly modified. It can be misunderstood now. As a minor remark, notice, that topology optimization algorithms are already successfully used in EM.

2/ Maybe I missed it, but I would like to see in the paper the mathematical relation, similar to (1.1)-(1.3) and (2.1)-(2.4), which has been used to evaluate the fitness function (according to the paper, it is a combination of gain and bandwidth metrics). I would expect something like

$f = w_1 * G + w_2 * B$, where G is antenna gain and B is fractional bandwidth.

I am especially interested in bandwidth optimization. How is it done? What are the weights? How is the bandwidth evaluated - from Q factor? From the reflection coefficient? Are the weights modified during the optimization? Are the weights also optimized?

3/ The simultaneous optimization of antenna gain and bandwidth should, in principle, deliver the Pareto front between gain and bandwidth. Can the authors comment on this and explain where their solution lies on this curve?

4/ The paper is relatively sparse with respect to the optimization's description. How do the authors code the GA genes with respect to the optimized parameters (position, length) - how long is the gene (how many binary optimization unknowns are used)?

Again, the paper is fine and can be published. Just add the technical details and correct some statements.

Reviewer #3 (Remarks to the Author):

I would like to thank the authors for answering the issues I raised. I think the paper is ready for publication now.

Reviewer #2 (Remarks to the Author):

The paper has been improved. There are, however, still some issues that might be corrected by the authors to make the manuscript better.

1/ The authors mixed the bounds (Harrington, Geyi), other bounds (Gustafsson, Capek, et al.), and inverse design (topology optimization). The bounds (Harrington, Geyi, Gustafsson, Capek, et al.) do not propose the actual shape of the antenna. They only predict what can, in principle, be the maximum value of such an antenna. Meanwhile, topology optimization/inverse optimization algorithms can deliver actual design. Notice that convex optimization routines used by Gustafsson and Capek also do not deliver an antenna, only establishing bounds, which is, as compared to Harrington/Geyi, the tight bound. Based on this, I believe the text on page 6, right column, must be slightly modified. It can be misunderstood now. As a minor remark, notice, that topology optimization algorithms are already successfully used in EM.

Our Response:

We agree on each point. We take the privilege to use the Referee formulation from here to emphasize this point in the revised version.

2/ Maybe I missed it, but I would like to see in the paper the mathematical relation, similar to (1.1)-(1.3) and (2.1)-(2.4), which has been used to evaluate the fitness function (according to the paper, it is a combination of gain and bandwidth metrics). I would expect something like

$f = w_1 * G + w_2 * B$, where G is antenna gain and B is fractional bandwidth.

I am especially interested in bandwidth optimization. How is it done? What are the weights? How is the bandwidth evaluated - from Q factor? From the reflection coefficient? Are the weights modified during the optimization? Are the weights also optimized?

Our Response:

Following the Referee's suggestion, we added more discussion to Fig 1, which demonstrates the algorithm steps. The objective function is built in steps – the first is to reach a threshold value in gain (the value is adjusted manually - to overcome Chu-Harrington and then improve it as much as possible). The second step is to assess the BW and check whether it overcomes a certain value (10% fractional bandwidth has been set as a target, relevant to most wireless communication needs).

This semi-manual approach was chosen after trying different gain-bandwidth combinations, like what the Referee suggested. While this form is quite appealing as it allows a higher level of automation and might lead to a better (close to optimal) solution, it didn't work. Our heuristic justification is that the highly resonant structure, which we explore, is very sensitive to changes (typical to superscattering and superdirective designs). Thus, putting too many constraints simultaneously leads to instability. Considering the time for the forward solver calculation, this approach didn't provide any reasonable results.

Following the Referee's recommendation, we introduced those explanations and an inline formula in the revised version.

The superdirective bandwidth was calculated as the range of frequencies for which the gain overcomes the Chu-Harrington or Geyi's limits (please, recall Fig. 1). This range of frequencies was checked to be continuous (gain does not drop below the limit within the range). We clarified this point in the revised.

Finally, the Referee suggests considering weight optimization. It is an extremely interesting and timely topic. It is also super challenging. Our new work (though a different topic) <https://arxiv.org/abs/2310.11199> considered meta-learning for those purposes. Here the fast computation of the forward problem is an absolute enabler. In the case of an antenna with radiation lobe calculation, this still seems challenging. Following the recommendation, we introduced this discussion too.

3/ The simultaneous optimization of antenna gain and bandwidth should, in principle, deliver the Pareto front between gain and bandwidth. Can the authors comment on this and explain where their solution lies on this curve?

Our Response:

The Referee keeps challenging us with extremely interesting and very deep questions that we didn't think of before – this is strongly appreciated. Pareto front (or frontier) is indeed a very relevant measure in multi-objective optimization. While computing the Pareto front is typically a hard problem, intuitively, gain and bandwidth are subject to an engineering trade-off. In the case of our step algorithm, we are not 100% sure that calculating the Pareto front is an entirely correct notion. We would like to keep this point in the discussion, as it is extremely interesting for future research – following the recommendation, we added the discussion in the conclusion of the revised version.

4/ The paper is relatively sparse with respect to the optimization's description. How do the authors code the GA genes with respect to the optimized parameters (position, length) - how long is the gene (how many binary optimization unknowns are used)?

Our Response:

Following the recommendations, we added more information in the revised version.

Again, the paper is fine and can be published. Just add the technical details and correct some statements.

Our Response:

We appreciate the Referee again – we learned quite a few from our interactions, we much value this and appreciate the Referee's time spent on our work.

REVIEWERS' COMMENTS:

Reviewer #2 (Remarks to the Author):

Thank you for the modifications of the manuscript. I think it is ready to be published.